# Topkapi: Parallel and Fast Sketches for Finding Top-K Frequent Elements

**Ankush Mandal**
School of Computer Science
Georgia Institute of Technology
Atlanta, GA
ankush@gatech.edu

**He Jiang**
Department of Computer Science
Rice University
Houston, TX
cary.jiang@rice.edu

**Anshumali Shrivastava**
Department of Computer Science
Rice University
Houston, TX
anshumali@rice.edu

**Vivek Sarkar**
School of Computer Science
Georgia Institute of Technology
Atlanta, GA
vsarkar@gatech.edu

## Abstract

Identifying the top-$K$ frequent items in a collection or data stream is one of the most common and important operations in large data processing systems. As a result, several solutions have been proposed to solve this problem approximately. We observe that the existing algorithms, although theoretically sound, are suboptimal from the performance perspective because of their limitations in exploiting parallelism in modern distributed compute settings. In particular, for identifying top-$K$ frequent items, *Count-Min Sketch* (*CMS*) has an excellent update time, but lacks the important property of *reducibility* which is needed for exploiting available massive data parallelism. On the other end, the popular *Frequent* algorithm (*FA*) leads to *reducible* summaries but its update costs are significant. In this paper, we present *Topkapi*, a fast and parallel algorithm for finding top-$K$ frequent items, which gives the best of both worlds, i.e., it is *reducible* and has fast update time similar to *CMS*. *Topkapi* possesses strong theoretical guarantees and leads to significant performance gains due to increased parallelism, relative to past work.

## 1 Introduction

Counting and identifying frequently occurring items, or "heavy hitters", is one of the most important and intuitive metrics to gain insight into large-scale data. The naive way to extract top-$K$ items from a data stream is to count the exact number of occurrences of each distinct item, then sort the histogram to obtain the most frequent items. This naive but popular approach suffers from a time complexity of $O(n \log n)$, in which $n$ is the total number of elements in the dataset, and also a space requirement of $O(n)$, assuming sorting is performed in linear space. In a distributed environment, where data sharding is common, the problem is quite severe. We would have to keep a local frequency histogram on each node, which is usually of size $n$ itself. These local histograms will need to be communicated across the nodes, and followed by global merge and sort operations. Thus, each node would need to communicate $O(n)$ sized histograms, which can lead to a significant communication bottleneck. Consider the simple task of keeping track of most popular phrases, of up to 4 words, on twitter feeds. With a vocabulary of over a million, the total number of items we need to keep track of becomes $n = (10^6)^4 = 10^{24}$. Similarly, counts of the number of clicks on "Amazon.com", given specific user's features and their combinations, in the past hour, are common in clickthrough prediction [12]. In general, the $O(n)$ time complexity becomes unacceptably large for "big data".

Fortunately, approximations often suffice in practice. Frequencies in most real word applications follow the Power Law [7], and therefore even approximately knowing the counts are enough to identify frequent items, also known as *heavy hitters*, efficiently. This feasibility for approximations allows for a significant reduction in computational and memory requirements. As a result, approximate counting is a very active and widely studied research area. There has been a remarkable success in obtaining algorithms for finding heavy hitters with exponential improvements in memory requirements, and a lot is known about the theoretical complexity of these algorithms [3]. Several of these algorithms are deployed in practice. Two notable algorithms include *Count-Min Sketch* (*CMS*) [7] which is hashing based and the *Frequent* algorithm (*FA*) [10] which is based on maps (or dictionaries).

However, even after 30 years of research on approximate counting over data streams, developing a practical algorithm that can fully utilize the massive amounts of available parallelism in the form of multi-core and multi-node (or distributed parallelism) is still an active area of research. Prior algorithms, such as [18], only rely on the theoretical reduction in communication, but require synchronized updates, for every increment, making them expensive in practice. In [2], the authors identify mergeable or *reducible* as a critical property that eliminates the need for synchronization. With the *reducibility* property, every node can create its summarization of the local data and transmit this exponentially small summary. Each of these little sketches can be merged to obtain the global summary of the data from which global heavy hitters can be identified.

It was argued in [2] that most popular algorithms, including *CMS*, are not suitable for the distributed setting because they lose the *reducibility* property, i.e., it is not possible to identify top-$K$ by merging local top-$K$ and their *CMS* summaries. Our experiments (section 5.2.8) confirm the significantly poor precision for *CMS* in distributed settings. Fortunately, the same paper [2] showed that *FA* is *reducible* and thus suitable for distributed computing. However, *FA* is costly to update; an update operation requires time that is linear in the size of the summary. Slow updates are also one of the main reason why *CMS*, despite being theoretically inferior, is preferred [7]. In contrast, *CMS* has only logarithmic update cost, which is desirable, but local *CMS* summaries cannot be combined (since they are not *reducible*). Thus, even if *CMS* is known to be faster than *FA*, it is not a suitable option in distributed setting.

To summarize, the popular hashing based *CMS* has logarithmic update cost but do not have the crucial *reducibility* property required for utilizing massive parallelism. On the other hand, non-hashing based *FA* summaries are reducible, but updates are significantly costly. In this paper, we show a theoretically sound and superior algorithm which combines both *CMS* and *FA* in a novel way that achieves the best-of-the-both worlds – logarithmic (efficient) updates as well as *reducibility* needed for parallelism. Our experiments show that the new proposal is on average 2.5x faster in practice than *FA* for distributed and multi-threaded execution.

**Our Contributions**    The problem addressed by this paper is to identify the top-$K$ frequent items in a given data stream(formal definition in sec 2.1). For this problem, we present *Topkapi*, a fast and parallel approximate algorithm. 1) *Topkapi* combines *CMS* and *FA* in a novel way that makes the summary *reducible* and at the same time capable of enabling parallelism. 2) We show that *Topkapi* retains the provable probabilistic error guarantees analogous to popular sketching algorithms in the literature. 3) We provide optimized parallel implementations for *FA*, *CMS* and our proposed *Topkapi* algorithm. Our implementation is optimized to overlap communication with computation and is capable of exploiting both multi-node and multi-core parallelism effectively. 4) We provide rigorous evaluations, profiling, and comparisons of two popular algorithms *CMS* and *FA* with *Topkapi* on large-scale word counting benchmarks. Our experiments indicate significant performance gains with *Topkapi* compared to existing approximate heavy hitters problem. 5) Our work also provides empirical quantifications of the benefits of using approximate algorithms over exact state-of-art distributed implementation in Spark. Our results show disruptive performance gains (sec 6 of *Supplementary document*), with *Topkapi*, over some of the fastest known exact implementations, at the cost of small approximations.

## 2   Background

### 2.1   Notations

We will refer the problem of finding the top-$K$ most frequent items in the data stream as the "top-$K$ problem". Let's assume we have $D$ distributed data streams $\{S_1...S_D\}$, for example, $D$ text streams. Let us assume that there are in total $M$ words $\{w_1...w_m\}$. Our goal here is to find $K$ most frequent words in these streams as an aggregate, i.e., $\cup_{i=0}^{D} S_i$ where the union represents concatenation (or

aggregation) of the streams. We represent the frequency of a word $w$ by $f$. Also, let $N$ denotes the summation of all the frequencies, i.e., $N = \sum f$. If the $K$-th most frequent word has frequency $f_K$, then we want to report all the words for which $f \geq f_K$.

Several approximate formulations of the *heavy hitter* problem were proposed to overcome the linear memory barrier. We use the standard formulation given in [2]. For details, refer to sec 1 of *Supplementary document*.

We will interchangeably use the word sketches and summary. They mean the same thing. Approximate algorithms for heavy-hitters produce a summary output which is typically much smaller than the data. This summary can be used to answer the heavy-hitters or other estimation queries.

Since we will be using approximate (lossy) algorithms over distributed clusters, where we will need to merge different summaries from different nodes, we need to define *reducibility* of the summaries (or sketches). *Reducibility* will ensure that the algorithm can be parallelized efficiently. Our definition of *reducibility* is inspired from the definition of mergeability in [2]. However, our definition is simpler and more generic for better readability.

**Reducible Summary:** Given the output summary $O_1$ from running algorithm $A$ on data stream $S_1$ and output summary $O_2$ with running the same $A$ on data $S_2$. We call an algorithm reducible if we can recover some summary $\hat{O}$ directly from the two output summaries $O_1$ and $O_2$, such that, if we use the combined summary $\hat{O}$ to replace $O$, which is a summary obtained after running $A$ on $S_1 \cup S_2$, we still retain all theoretical guarantees of algorithm $A$. In addition, we want two more conditions: – 1) The computation cost of calculating $\hat{O}$ from $O_1$ and $O_2$ should be less than the cost of running $A$ over $S_1 \cup S_2$ and 2) The space required by $\hat{O}$ should not be more than that of $O$.

Note that sometimes the algorithm $A$, such as *FA* (defined later), is sensitive to the order in which it sees the input data. In such cases, we cannot guarantee that the combined summary $O$ will be equal to $\hat{O}$, but so long as the final outputs have same accuracy guarantees and computation time, we can distribute it efficiently.

## 2.2 Exact Algorithms

Exactly solving the top-$K$ problem requires $O(M)$ memory and have $O(MlogM)$ runtime complexity. One can compute all the frequencies $f$ using standard word count or histogram computation. Then sort the words based on the frequencies $f$ as the key and report the top-$K$ words. We can utilize hash-maps to store words and update frequencies as we read the data. Finally, we sort the map.

A unique advantage of this exact method is that it is easy to parallelize. We can perform separate hash map updates with separate data in parallel, and at the end, we perform reduction by key to get the final frequencies. Then we sort the words to get the top-$K$ frequent words. Several state-of-the-art implementations, such as Spark based `wordcount() + sort()` use this method. However, our experiments (sec 6 of *Supplementary document*) reveal that $O(M)$ storage and communication, even with the best possible distributed implementation can be orders of magnitude slower compared to approximate solutions in a distributed setting.

## 2.3 Approximate Algorithms

Algorithms for finding approximate heavy hitters is a heavily studied topic in database and theory community. These algorithms mainly come in two flavors - 1) counter-based and 2) sketch-based.

**Counter-based Algorithms:** Counter-based algorithms maintain a set of counters (maps) associated with a subset of words (or maps with counters) from the data stream it has traversed. This subset of words is called the monitored set. There are several variants, such as Frequent [10], Lossy Counting [11], and Space Saving [13]. Please see [6] for a good survey on them. Note that, [6] explored only sequential version of these algorithms whereas we are mainly interested in parallelism here. In our work, for comparison with counter-based approach in general from the perspective of parallelism, we consider one of the most popular variant – *Frequent Items* or simply *Frequent* algorithm (*FA*) (a brief description of important features is given in sec 2 of *Supplementary document*). The main advantage of this approach is the summaries are *reducible* whereas the main disadvantage is high update time.

**Sketch-based Algorithms:** Instead of maintaining counters for a monitored set of words, sketch-based algorithms use lossy hashes to create a summary which can be used to estimate the frequency of

any given item. For this study, we consider one of the most popular and efficient among the sketching algorithms – *Count-Min Sketch* (*CMS*), which is widely adopted in practice. Important algorithmic aspects of *CMS* are described in sec 3 of *Supplementary document*. Sketch-based approach provides fast update of summary but has significant disadvantage when it comes to *reducibility* because heap, which is not reducible, is needed for recovering identity of counters.

Although there has been a significant development in past years on approximate heavy hitters [3; 14; 7; 11; 8], little focus has been given on the parallelism aspects except a very few, such as [17; 4; 5; 15]. When it comes to parallelism, there are several choices. Parallelizing the individual updates is not a good option as the computation is too low to justify parallelism. Exploiting parallelism just for one update is too fine-grained, and the overhead of parallelism would be much higher than the gain from parallelism. Data parallelism, i.e., performing computation for different blocks of data in parallel, is more preferred because we have a much better granularity of parallelism. Thus, with enough data, it is always preferred to have each parallel process work on its own memory and later a one-time merge. We also get a very high degree of parallelism due to the large size of the data. Thus, it is essential for the algorithm to be *reducible*. However, with data parallelism, the algorithmic update time becomes a factor with a significant impact on performance. [17; 4; 5] discuss parallel counter-based Space Saving [13] algorithm over CPU, GPU, and distributed environment respectively. However, none of them addresses distributed environment with multi-threading. Also, we can see in [4] that the counter-based approach has significant update time even on massively-parallel architecture such as GPU. Interestingly, [15] explored fine grain parallelism to speedup Space Saving on modern CPUs with advanced vector instructions. This kind of exploitation of fine grain parallelism is complementary to coarse grain parallelism which is the main focus of this work.

## 3   Our Proposal: Topkapi

### 3.1   Intuition

Consider the *CMS* matrix $M$ (sec 3 of *Supplementary document*) without the overhead of updating the heap for identifiability. Note that every row of this matrix is a simple hashed counter, and all rows are independent. Thus, without the heaps, *CMS* are *reducible* summaries, i.e., different summaries with the same hash functions can be merged by simply adding the sketches. The update time is mere $\log \frac{1}{\delta}$ ($\delta$ is failure probability) which is also the number of independent hash functions needed. Following [16], in all our experiments, only 4 hash functions suffice in practice. An important observation is that the sketch matrix $M$ is enough to estimate the counts of any given item accurately but cannot identify the frequent items on its own. Thus, without identifiability, we need another pass over every item, estimate its count, and then report top-$K$. Given the number of unique items is astronomical, this is prohibitive. However, if we can somehow efficiently identify a small enough set of candidates $CS$ which likely contains the most frequent elements then we just have to check every element in $CS$, instead of all the items.

It should be noted that due to simple hashing, every cell of *CMS* will count the total occurrence of a small set of items ($\epsilon N$ in expectation). $\epsilon$ is approximation parameter. If a heavy hitter item $HH$ with $f \geq \phi \times N$ hashes to this counter, it is very likely to be the most frequent item in the cell. Thus, if we can identify the heaviest element in the subset of stream in every cell efficiently, then there is hope of getting a good enough candidate set $CS$.

*FA* keeps the identity of the heavy hitters in a map. The update time is equal to the size of the map, which needs to be $\frac{1}{\epsilon}$ for reporting all the heavy hitters. However, if we are interested in just the heaviest item, then we don't need maps and the update time will be constant. We just need two cells; one stores the identity of the heaviest element and another a counter to increment/decrement.

The above observations form the basis of our proposal. We propose to associate a *FA* summary of size 1 to each counter of *CMS*. We later show that it has sound theoretical guarantees analogous to *CMS* for solving approximate heavy hitters problem. Furthermore, this modification eliminates all the issues mentioned in section 2.3.

### 3.2   Topkapi: Algorithm Descriptions

*Topkapi* contains a *CMS* summary, i.e., a two-dimensional array $l \times b$ $M$. As a reminder, $b$ represents number of buckets for a hash function and $l$ represents the number of hash functions. We have $l$ pair-wise independent hash functions $h_1, h_2, ..., h_l$ to map words to the range $\{1, 2, ..., b\}$. $b$ is set to $(\frac{1}{\epsilon})$ and $l$ is set to $log\frac{2}{\delta}$. Now, each cell $M_{i,j}$ has in addition two more components: - 1) $LHHcount_{ij}$ representing the count of frequent item associated with $M_{ij}$ (Local Heavy Hitter

count) and 2) $LHH_{ij}$ containing the word (identity) whose frequency is stored in the $LHHcount_{ij}$. This $LHH_{ij}$ will ideally be the most frequent item mapping to $M_{ij}$. Note, each item is mapped to $l$ cells in $M$.

During initialization, all the $LHHcounts$ as well as $M$ are set to 0. During processing of data stream, we do the usual update of $M$, the *CMS*. In addition, for each word $w$, we compare $w$ with the $LHH$ of the cell at $h_i(w)$. If it matches, then we increment the corresponding $LHHcount$ of the cell at $h_i(w)$. Otherwise, we decrement the $LHHcount$. If the decrement causes the $LHHcount$ to become 0, then we replace the $LHH$ of $h_i(w)$ with $w$ and set the corresponding $LHHcount$ to 1. We do this $\forall i : 1 \le i \le l$.

In the end, we consider the union of all the unique $LHH$ values as the candidate set $CS$. We estimate their counts using the *CMS* and finally report all elements with the count higher than some threshold like $\phi \times N$ for $\phi$-heavy hitters problem.

### 3.3  Topkapi: Properties

Here, we summarize the main algorithmic properties of *Topkapi*. For detailed theoretical analysis of *Topkapi*, please see sec 4 of *Supplementary document*. An important thing to note here is that we do not require any heap for *Topkapi*.

1. *Topkapi* with size $l = \log\lceil\frac{2}{\delta}\rceil$ and $b = \frac{1}{\epsilon}$ solves the $\phi$-approximate heavy hitter problem provided ($\epsilon < \phi$).

2. *Topkapi* data structure is reducible. As a result, *Topkapi* can exploit parallelism easily.

3. *Topkapi* data structure has update cost of $\log\frac{2}{\delta}$ which is similar to logarithmic update cost of *CMS*.

It is noteworthy to mention that if we want to get the frequency estimates along with the identities of top-$K$ frequent elements, we can use both *CMS* count (overestimates) and *LHH* count (underestimates) to take an average and decrease the error constants, else we can always use the estimate from *CMS*. So, we are strictly better.

### 3.4  Practical Considerations

In *Topkapi*, the only use of *CMS* counters in $M$ is estimation. It turns out that in practice $LHHcount$ itself is also a good estimator of the true frequency of $LHH$. This is because we are using *FA* summary of size 1 on a tiny stream. Thus, if our goal is only to get the identities of top-$K$ frequent elements, we can altogether get rid of *CMS* counters and reduce the memory overhead significantly.

Finally, towards the end, instead of considering all the unique $LHH$s, we can be little smarter. Note that every item is mapped to every row and all the rows are independent. The idea is to perform a linear scan over only the 1st array ($l = 1$) of counters and add $LHH$ into $CS$ if the corresponding $LHH$ is greater than a threshold in any of the $l$ rows. Then we sort the candidate set $CS$ to identify top-$K$ candidates according to their $LHHcounts$ and report the $LHH$s associated with highest $LHHcounts$. Pseudocode of this practical version of *Topkapi* is given in Algorithm 1. We will use this algorithm in experiments.

## 4  Implementation

It is imperative that we use multi-core parallelism along with distributed parallelism to make effective use of current and future computing systems.

### 4.1  Multi-core Parallelism

When considering intra-node parallelism using multi-threaded execution, we have several options for *Topkapi*. We can use different threads for different hash functions in $\{h_1, h_2, ..., h_l\}$. However, this limits the number of threads to the number of hash functions which is usually quite low. Another option is to use different threads to process different chunks of data and use a single sketch shared across different threads. The threads will then have to use locks or atomic variables to perform the shared update of counters in the sketch. The use of locks or atomic variables can create significant contention due to the distribution of word frequencies. As the heavy hitters are most frequent, it is highly likely that many threads encounter the same heavy hitter word and try to update the same counter in the sketch.

**Algorithm 1:** Topkapi

**Data**: Input text stream $S$, parameter $K$

**Result**: top-$K$ frequent words in $HH$

1   $b \longleftarrow \lceil \frac{1}{\epsilon} \rceil$

2   $l \longleftarrow log \frac{2}{\delta}$

3   $C \longleftarrow l \times b$ counters

4   $C[i][j].LHHcount \longleftarrow 0 \quad \forall i \in \{1, 2, .., l\} \ and \ \forall j \in \{1, 2, ..., b\}$

5   **for** $w \in stream \ S$ **do**

6     **for** $i \in 1, 2, ..., l$ **do**

7       *calculate* $h_i(w)$

8       **if** $C[i][h_i(w)].LHH == w$ **then**

9         $C[i][h_i(w)].LHHcount \longleftarrow C[i][h_i(w)].LHHcount + 1$

10       **else**

11         $C[i][h_i(w)].LHHcount \longleftarrow C[i][h_i(w)].LHHcount - 1$

12         **if** $C[i][h_i(w)].LHHcount == 0$ **then**

13           $C[i][h_i(w)].LHH \longleftarrow w$

14           $C[i][h_i(w)].LHHcount \longleftarrow 1$

15   **for** $j \in 1, 2, ..., b$ **do**

16     **if** $C[1][j].LHH \ OR \ C[i][h_i(C[1][j].LHH)].LHH > Threshold \ \forall i \in \{2, .., l\}$ **then**

17       $CS.\texttt{insert}(C[1][j])$

18   `sort`$(CS)$ *in descending order of* $LHHcount$

19   `report` $LHH$ *of* $CS$ *entries with top K highest* $LHHcount$

---

We can mitigate the problems mentioned in the previous options by exploiting high level of data parallelism at the cost of extra local memory. We can create thread-local copies of sketches and use different threads to process different chunks of data. Then we exploit the reducibility property of the sketch and merge the thread-local sketches at the end of the data traversal to produce a single sketch for a node. We observe that even for a large dataset, we only need a small sketch. For example, with $l = 4$ and $b = 1024$, the size of the *count* array is 16KB and the size of the *id* array is 64KB. So, the amount of extra memory required is quite low. As different threads are working on their own local copies of the sketch, we do not need locks to update a counter anymore.

### 4.2 Distributed Parallelism

Since our algorithm is *reducible*, distributed parallelism is quite straightforward. We start with multi-threaded execution of *Topkapi* on each node following the method mentioned in section 4.1. When we have the final summaries ready at each node, we perform a parallel reduction or merging of the summaries to get a final summary at the root node. Once we have that, we use the final summary at the root node to perform the potential top-$K$ candidate set ($CS$) construction, `sort` $CS$, and `report` top-$K$ words steps from the sequential *Topkapi* pseudocode mentioned in Algorithm 1.

**Communication cost -** One important factor considering distributed computation is the communication overhead. The communication traffic for merging summaries between two nodes is the size of a single summary. As we use a parallel reduction strategy to merge the summaries at different nodes, we perform $logD$ such merging steps between different pairs of nodes, where $D$ is the total number of nodes.

**Overlapping Communication with Computation -** In distributed computing, one can hide some of the communication overhead by carefully coordinating the communication so that it overlaps with the computation. In our implementations, we also exploit such opportunities. The reduction algorithm merges all the counters of a summary independently, i.e., a merged counter only depends on the respective two counters from the two summaries being merged. Hence, we can overlap the communication for a specific row of $b$ counters with the computation of merging the previous rows of $b$ counters. We use MPI non-blocking communication to achieve this overlapping.

For an overview of distributed and multi-threaded implementation of *Topkapi*, we present the pseudocode in Algorithm 2 which extends the pseudocode from Algorithm 1.

### 4.3 Parallelizing Baselines: Frequent Algorithms and Count-min Sketch

For the purpose of performance comparison, we choose the two most popular algorithms, namely "*FA*" and "*CMS*" as representatives from counter-based algorithms and sketch-based algorithms respectively.

As mentioned in section 2.3, *CMS* requires a heap for finding top-$K$ and is not *reducible*. Due to this exact reason, [2] instead used *FA* for mergeability. Unfortunately, without *reducibility*, it is hard to exploit massive data parallelism independently, and the implementations are unlikely to be efficient.

We made a simplifying assumption that each subsample of the stream is uniformly distributed and hence merging two top-$K$ still make sense.

---
**Algorithm 2:** Topkapi_Parallel(S[][], K, N, T)
---
1 **for** $i \in nodes\ N$ **do**
2     **for** $j \in threads\ T$ **do**
3        *create thread local copies of Topkapi summary*;
4        *execute* Topkapi *for data S[i][j] in parallel using summary j with only the summary update phases*;
5     merge *thread local* $summary_j$ $\forall j \in \{1,...,T\}$ *to produce node final* $summary_i$;
6 *use* parallel reduction *strategy to* merge *node final* $summary_i$ $\forall i \in \{1,...,N\}$ *to produce a final summary at root node*;
7 construct *CS using final summary at root node*;
8 sort *CS and* report *top-K words from root node*
---

There were two main quest behind making this dumb assumption with *CMS*. **1) Does Reducibility Matters in Practice?** Subsampling streams is one of the most popular ways of reducing computation. The assumption is that the frequent item in the whole stream is also a frequent item in any small subsample of the stream. If this holds, then merging top-$K$ across substreams should be possible and reducibility may not matter much in practice for accuracy. We aimed to check this hypothesis. **2) In the most lucky world, is CMS still the fastest?** *CMS*, even with heaps, has significantly faster update time compared to *FA* (experimental results in Figure 1f). Can *Topkapi* beat this cheap *CMS* variant on performance?

Thus, to understand the performance benefits, we ignored the accuracy aspect and merged the heaps. To merge the heaps, we perform naive merge where we take two heaps and sort them to make a final heap containing top-$K$ candidates. One can argue that increasing the heap size (e.g., $2K$) would improve the accuracy of *CMS*. So, we give *CMS* more room to get better accuracy by using a heap size of $4K$. It should be noted that only the sketch (counters) in *CMS* is *reducible* and the reduction is performed similarly as *Topkapi*.

## 5 Evaluations

### 5.1 Code and Experimental Setup

The implementations of our algorithm[1] and competing algorithms are in C++ under a common framework to ensure as much of an apples-to-apples comparison as possible when presenting relative performance results. As for data we have used text data from the Project Gutenberg [1] corpus and PUMA Datasets [9]. The details on experimental setup and datasets are given in sec 5 of *Supplementary document*. For all the experiments, $K$ is set to 100 unless otherwise stated.

### 5.2 Results

#### 5.2.1 Scalability over Number of Nodes

We present strong scaling (fixed data size) performance results over varying number of nodes for two different data sizes: a) 16GB (Gutenberg dataset) and b) 128GB (Puma dataset). Figure 1a and Figure 1b represents the speedup of *Topkapi* over *Frequent*(*FA*) and *Count-Min Sketch*(*CMS*) for 16GB and 128GB data sizes respectively for 1 to 16 nodes with each node running 8 threads. We see that our proposal consistently get roughly 2.5x speedup over *FA* for both the data types whereas we usually get sightly lower speedup over *CMS*. It should be noted that we used the dumb merging of top-$K$ heap for *CMS* which loses significant accuracy (see Section 5.2.8). Despite this cheap approximation with *CMS*, we still observe 2x-2.6x speedup for 16GB data and 1.6x-2x speedup for 128GB data over *CMS*.

#### 5.2.2 Scalability over Number of Threads

Figure 1c represents the performance improvement of *Topkapi* over *FA* and *CMS* for 1 to 64 threads on a single node with 32 cores. We used 16GB data for this experiment. The plot shows that we get around 2x speedup over *CMS* for all the data points whereas we get similar performance improvement over *FA* till 8 threads; after that speedup over *FA* increases steeply and we get 22x speedup with 64 threads. As an optimized implementation of *FA* requires two hash-maps with size being in the order of number of counters, the memory footprint of *FA* is quite high. This negatively affects the performance after a threshold when L3 cache can not contain all the data footprint of two or more threads in the same processor chip. This performance degradation becomes more pronounced when

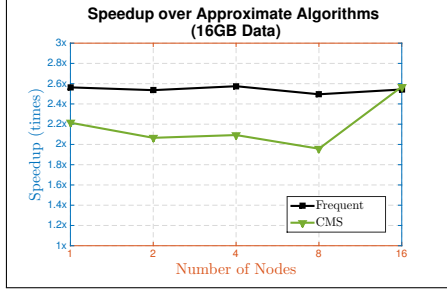

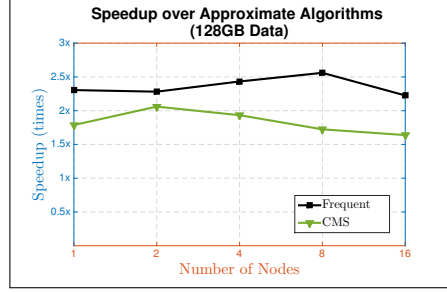

(a) Performance comparison with *FA* and *CMS* for 16GB data. Number of threads per node is 8. Used a cluster of Intel®Westmere processors with each node having 12 cores.

(b) Performance comparison with *FA* and *CMS* for 128GB data. Number of threads per node is 8. Used a cluster of Intel®Westmere processors with each node having 12 cores.

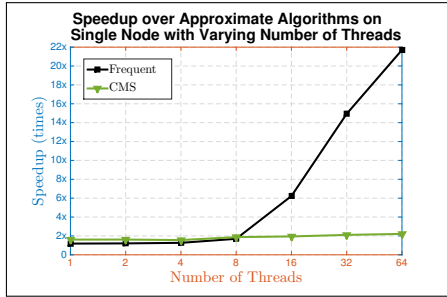

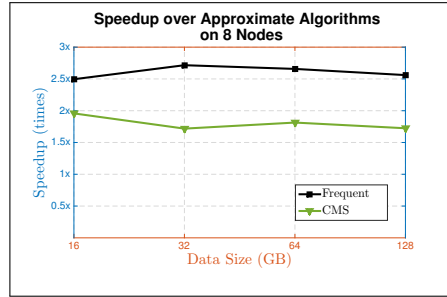

(c) Performance comparison with *FA* and *CMS* for varying number of threads. Data Size=16GB and Number of Nodes=1. Used a single node with 32 cores from four IBM Power®7 chip.

(d) Performance comparison with *FA* and *CMS* for varying data size on 8 nodes. Number of threads per node is 8. Used a cluster of Intel®Westmere processors with each node having 12 cores.

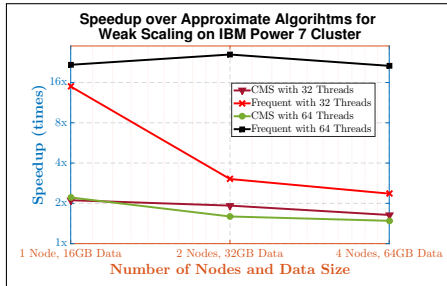

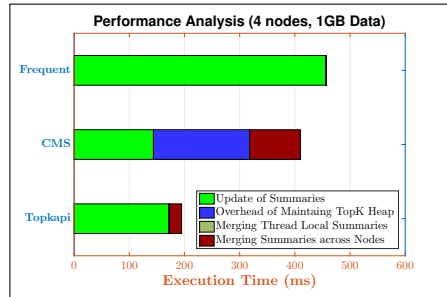

(e) Performance comparison with *FA* and *CMS* for high number of threads (32 and 64) in distributed setting. Used a cluster of IBM Power®7 processors where each node has 32 cores from four processor chips.

(f) Execution time break down for *Topkapi*, *FA*, and *CMS* for 4 nodes and 1GB data size. Number of threads per node is 8. Used a cluster of Intel®Westmere processors with each node having 12 cores.

Figure 1: Performance Results

more than one hardware thread is executed on the same core. For example, the configuration with 64 threads uses the SMT feature of Power®7 and executes 2 threads on each core.

### 5.2.3 Scalability over Data Size

To see the effects of data size on performance, we fix the number of nodes to 8 and vary the data size from 16GB to 128GB. The resulting plot with speedup over *FA* and *CMS* is given in Figure 1d. The figure represents around 2.5x speedup over *FA*, and 1.5x-2x speedup over *CMS*. Beside these good performance improvements, the consistency of speedup indicates that *Topkapi* performs well for a wide range of data sizes.

### 5.2.4 Scaling over Number of Nodes with Increasing Data Size

Now, we increase the data size along with the number of nodes and use high number of threads (32 and 64 threads) on each node to find out how we perform in terms of weak scaling. Figure 1e presents the resulting plot. As we can find from the plot, we get consistent speedup of roughly 2x for

Table 1: Precision Comparison between Approximate Methods

| Data Size | Precision(%) | | | |
|---|---|---|---|---|
| | *Topkapi* (1024 Counters) | *CMS* (1024 Counters) | *CMS* (2048 Counters) | *FA* (1024 Counters) |
| 16GB | 96 | 64.4 | 68.33 | 87 |
| 128GB | 95 | 11.6 | 49.66 | 94 |

*CMS*. However, we see some interesting pattern for *FA*. For 32 threads, the speedup over *FA* decreases significantly as move from one node to 2 nodes setting. On the other hand, the speedup remains high (more than 16x) for 64 threads through out all data points. In case of *FA*, the merging of summaries has lower computational overhead compared to *CMS* and *Topkapi*. So, when we move to distributed setting with 2 or more nodes, it boils down to which factor has more impact - the performance gain from low overhead merging step or the performance degradation from high level of multi-threading.

### 5.2.5 Performance Analysis
Figure 1f represents the performance break down of *Topkapi*, *FA*, and *CMS* execution. The plot supports our analysis that *FA*, among all three algorithms, has the highest update time for the summary but lowest cost when it comes to merging summaries across nodes. Undoubtedly, *CMS* has lowest update time for the summary because it involves only calculating the bucket through hashing and then incrementing the respective counter. However, its performance for "top-$K$ problem" is highly thwarted by the overhead of maintaining probable top-$K$ words summary. So, the effective update time for *CMS* becomes quite high. While *Topkapi* needs a slightly higher update time than *CMS*, its effective update time is much lower because it does not involve any overhead from maintaining heap. Furthermore, *Topkapi* has quite low computational cost for merging summaries across nodes whereas *CMS* has the highest cost in this regard.

### 5.2.6 Performance over Varying *K*

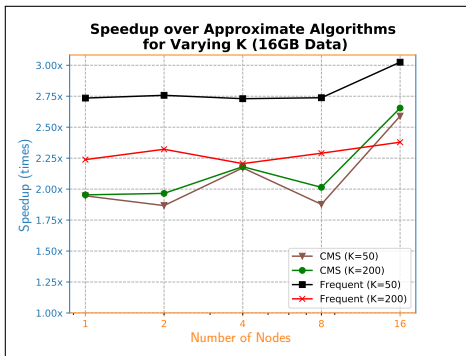

Figure 2: Performance comparison with *FA* and *CMS* for $K$=50, 200 on 16GB data. Number of threads per node is 8.

We carried out the experiments related to Figure 1a for $K$=50 and $K$=200, and represented the results in Figure 2. We used 512 and 2048 buckets or counters respectively for $K$=50 and $K$=200. Speedup of *Topkapi* over *FA*, for $K$=50, increases to the range 2.73x-3.01x and for $K$=200, it decreases to 2.21x-2.36x compared to $K$=100. However, the speedup over *CMS* remained almost the same. When $K$ is smaller, *FA* should slow down since it now has a lesser number of counters ($1/\epsilon$ or $O(K)$) or tracked elements. So, it will more frequently perform the computation related to element not found, which is costly. For the same reason, *FA* will be faster when $K$ is larger. For each match, it only has to increment the corresponding counter, which is cheap. On the other hand, we do not expect the performance of *Topkapi* and *CMS* to change much apart from slight slowdown with increasing sketch size.

### 5.2.7 Comparing *CMS* with Separate top-*K* Pass
In batch processing environment, one may employ a two-pass algorithm where the first pass consists of pure *CMS* to get frequency estimates and a separate second pass for hash-based top-$K$ identification. In our experiments using 1 to 16 nodes (8 threads on each node) with 16GB data, we find that the execution time of this two-pass algorithm is on an average 0.97x of single-pass *CMS*+heap based approach. It is noteworthy to mention that the comparison is not fair since in a streaming setting, remembering the items itself, for the second pass, is of linear cost which is prohibitive.

### 5.2.8 Precision for Reported top-*K*
As *Topkapi* is *reducible*, it is expected to give good precision and Table 1 shows us exactly the same thing. *Topkapi* outperforms *CMS* and *FA* for precision over 16GB and 128GB data. Moreover, the poor precision observed for *CMS* indicates that the simplification we assumed in section 4.3 to favor better performance for *CMS* does not hold true.

**Acknowledgments**

This work was supported in part by NSF-1629459, NSF-1652131, NSF-1838177, AFOSR-YIP FA9550-18-1-0152, BRC grant for Randomized Numerical Linear Algebra, Amazon Research Award, and Data Analysis and Visualization Cyberinfrastructure funded by NSF under grant OCI-0959097 and Rice University.

## Footnotes

[1]https://github.com/ankushmandal/topkapi.git

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
