[Supplementary Material]

# Supplementary Material for "Topkapi: Parallel and Fast Sketches for Finding Top-K Frequent Elements"

**Ankush Mandal**
School of Computer Science
Georgia Institute of Technology
Atlanta, GA
ankush@gatech.edu

**He Jiang**
Department of Computer Science
Rice University
Houston, TX
cary.jiang@rice.edu

**Anshumali Shrivastava**
Department of Computer Science
Rice University
Houston, TX
anshumali@rice.edu

**Vivek Sarkar**
School of Computer Science
Georgia Institute of Technology
Atlanta, GA
vsarkar@gatech.edu

## 1   $\phi$-Approximate Heavy Hitters

Given an approximation parameter $\epsilon$, the approximate heavy hitters solution returns a set of words (items) $HH$ that satisfies the following two conditions with high probability ($\geq 1 - \delta$) - a) All words $w$ having $f > \phi \times N$ is present in the returned set $HH$ and b) every word in the set $HH$ is guaranteed to have $f > (\phi - \epsilon)N$.

We collectively call the algorithms solving this approximation as "approximate algorithms". Approximation breaks the linear complexity barrier and allows us to work with only logarithmic memory, with an insignificant loss in accuracy.

## 2   Frequent Algorithm

In 1982, Misra and Gries [11] first proposed a generalization of *Majority* algorithm (finds the most frequent element) to extend it for "top-$K$ problem". The same algorithm was rediscovered in 2002 by Demain *et al.* [8] and Karp *et al.* [10]. We refer to these algorithms by the general term "*Frequent*" algorithm (*FA*). *FA* keeps $(1/\phi)$ number of counters for finding all words with $f > \phi \times N$. During stream traversal, each new word is compared against the monitored set. If the element exists in the monitored set, then its count is incremented. Else, if there is some non-allocated counter, i.e., counter with count zero, then allocate the counter for the new item and set its count to 1. If all counters are already allocated, decrement all counters. In this process, if the count of any counter becomes 0, declare the counter as non-allocated and remove the associated word from the monitored set. As observed by Bose *et al.* [5], setting the number of counter to $(1/\epsilon)$ for *FA* solves the approximate frequency estimation problem. This algorithm is deterministic and achieves optimal theoretical guarantees.

The algorithm requires maintaining a map from strings to integer of size $(1/\epsilon)$. We briefly highlight three important aspects of this algorithm which will be used to contrast it with other algorithms

1. $(1/\epsilon)$ **per Update** The update cost of addition is $(1/\epsilon)$ in the worst case as we have to decrement counters.

2. **Reducible** It was shown in [2] that maps used in *FA* is reducible, and hence can be easily parallelized across multiple nodes.

3. **Map Overheads** To identify the items exactly, we need a map of strings to counters. Addition to maps creates additional overheads of resolving the hash collisions [12].

# 3 Count-Min Sketch

The Count-Min Sketch (*CMS*) algorithm proposed by Cormode and Muthukrishnan [7] is inspired by widely popular data structure called *Bloom Filter*[4] which is used for estimating counts of items over data stream while using high level of compression. The sketch is a two-dimensional array $M$ of $l \times b$ counters. Here we use $l$ 2-universal hash functions $h_1, h_2, ..., h_l$ which map words to $\{1, 2, ..., b\}$. These hash functions are pair-wise independent. For each occurrence of word $w$ in data stream, we increment the counter $h_i(w)$, in the $i^{th}$ row for $\forall i : 1 \leq i \leq l$. Any query of frequency estimation $\hat{f}$ of any word $w$ returns $min\{h_i(w) \forall i : 1 \leq i \leq l\}$. [7] proved that the expected error in frequency estimation is always an overestimate $\leq \left(\frac{N}{b}\right)$ and using $l$ hash functions reduces the error exponentially with $l$. So, using $l = O(log\frac{1}{\delta})$ and $b = O(\frac{1}{\epsilon})$ ensures the error in frequency estimation is $\leq \epsilon N$ with probability $1 - \delta$.

Since the algorithm only needs lossy hash functions, it does not require a map and can work with arrays. However, as the hash functions are not invertible, sketch-based methods do not preserve the identity of words associated with specific counters. Thus, to identify heavy hitters, we need additional data structures. There are two workarounds – 1) Dyadic interval trick [6] and 2) use of Heaps. The dyadic interval trick requires a tree of individual count-sketches and the memory overhead of tree is prohibitive in practice. The common workaround is to maintain a heap of top-$K$ words along with the sketch while reading the data stream. We will focus on this practical variant.

The sketch keeps track of the number of words processed so far ($n$). For each word $w$ in the data stream, we first update the sketch and then query the frequency estimation $\hat{f}$ of that word. If $\hat{f} \geq \phi \times n$, we search the word in heap. If the word already exists in a heap, then we update its count. Otherwise, we insert the word to the heap. If the heap is already full, we check if $\hat{f}$ is greater than the min count in a heap. If so, we do delete-min on the heap and insert the word.

*Count-Min Sketch* has the following key properties

1. $\max\left(\log\frac{1}{\delta}, logK\right)$ **per Update:** The update cost only requires adding to $\log\frac{1}{\delta}$ counters. If we need to update the heap, it requires additional $\log K$ operation. The total cost is logarithmic and hence significantly smaller than $(1/\epsilon)$ in practice.

2. **Not Reducible:** Since only the identities of top-$K$ items are stored in a heap, we cannot merge top-$K$ over two different streams to obtain the global top-$K$.

3. **Heap Overheads:** Although, the sketch only consists of arrays, and 2-universal hash functions are cheap, to identify top-$K$ items we have to use the heap data.

# 4 Topkapi: Theoretical Analysis

Before we argue about *Topkapi*, we review one useful known theoretical fact about *CMS* which we will use in the proofs.

**Theorem 4.1** *For every $w$ with frequency $f$ and its estimate $\hat{f}$ using CMS of size $l = \log[\frac{1}{\delta}]$ and $b = \frac{1}{\epsilon}$, we have the following with probability $1 - \delta$*

$$f \leq \hat{f}^{CMS} \leq f + \epsilon N \tag{1}$$

Note, we need $l = \log[\frac{1}{\delta}]$ to ensure the above for all $N$ after union bound.

Using the theorem above, we can show the following for *Topkapi*.

**Theorem 4.2** *Topkapi with size $l = \log[\frac{2}{\delta}]$ and $b = \frac{1}{\epsilon}$ solves the $\phi$-approximate heavy hitter problem provided ($\epsilon < \phi$) (Definition in section 1)*

**Proof:** Follows from two lemmas below combined with the definition of approximate heavy hitters instance.

**Lemma 4.3** *Topkapi with $l = \log[\frac{2}{\delta}]$ and $b = \frac{1}{\epsilon}$ ($\epsilon < \phi$) misses to report $w$ with $f \geq \phi \times N$ with probability at most $\frac{\delta}{2}$*

**Proof:** $w$ is missed if it is not in $h_i(w).LHH$ $\forall i$. For any $i$, $h_i(w).LHH \neq w$ implies that the *CMS* counter for $h_i(w)$ given by $M_{i,h_i(w)}.CMScounter \geq 2f$, otherwise local *FA* summary will not miss $w$. Thus, $w$ is not reported by any of the $i$ rows implies $h_i(w).CMScounter \geq 2f$ $\forall i$. Since the *CMS* estimate is the minimum of all $i$ rows, it means the estimate of *CMS* is at least $2f$ or $\hat{f}^{CMS} > f + f > f + \epsilon N$ which happens with probability at most $\frac{\delta}{2}$ from Theorem 4.1.

**Lemma 4.4** *Topkapi with $l = \log[\frac{2}{\delta}]$ and $b = \frac{1}{\epsilon}$ reports $w$ with $f \leq (\phi - \epsilon) \times N$ with probability at most $\frac{\delta}{2}$.*

**Proof:** We report $w$ only when its estimate $\hat{f}^{CMS} \geq \phi N$. Thus, if we report $w$ and $f \leq (\phi - \epsilon) \times N$, it implies that $\hat{f}^{CMS} \geq \phi N \geq f + \epsilon N$. Thus, the error of *CMS* estimate exceeds $\epsilon N$ which happens with probability at most $\frac{\delta}{2}$ from Theorem 4.1.

The following is immediately clear from the description of the algorithm

**Theorem 4.5** *Topkapi data structure has update cost of $\log \frac{2}{\delta}$.*

Finally, we can easily show that *Topkapi* is reducible

**Theorem 4.6** *Topkapi data structure is reducible.*

**Proof:** The counters in *CMS* is reducible, and furthermore, *FA* is reducible. The proof follows from the fact that every cell of *Topkapi* (*CMS* counter and *FA* of size 1) is reducible.

# 5   Experimental Setup and Datasets

The implementations of our algorithm and competing algorithms are in C++ under a common framework to ensure as much of an apples-to-apples comparison as possible when presenting relative performance results. We would like to mention that we have used a heap size of **4K** for *CMS* to allow better accuracy since the heap containing top-$K$ frequent words lacks the *reducibility* property. We used MurmurHash3 [3] for hash functions in all of the implementations to maintain comparability across different algorithms.
We compiled all codes using GCC 6.2.0 with the following flags: a) GNU C++11 extension, b) "O3" optimization flag, and c) OpenMP flag because we used OpenMP for multi-threading inside a node. We also used Boost 1.64.0 and OpenMPI 1.10.3 libraries for our code. To evaluate performance scalability for multi-node distributed computing with multi-threaded execution on each node, we ran many of our experiments on cluster of Intel®Westmere nodes with 12 processor cores per node running at 2.83 GHz. All of these nodes are connected via QDR InfiniBand (40 Gb/s) to each other. We used 8 threads per node for all of these experiments. Further, to show performance scalability in executions with large numbers of threads, we ran our experiments on a cluster of IBM POWER7®(P750) processors with 32 cores per node running at 3.8 GHz. IBM POWER7®processor supports 4-way SMT (simultaneous multi-threading) which let us launch up to 128 hardware threads per node.

## 5.1   Performance Metrics

Here, we define the performance metrics used in our work and also in past work:
**Precision -** This metric "Precision" here represents the ratio of number of correct top-$K$ frequent words reported to the total number of words reported.
**Recall -** "Recall" is ratio of correct top-$K$ frequent words reported to the value of $K$, K, regardless of the total number of words reported in the denominator of the Precision ratio. (Note that both the Precision and Recall ratios have the same numerator.)
**Speedup -** When we say performance "Speedup", we refer to the following ratio:

$$\frac{execution\ time\ of\ referred\ algorithm}{execution\ time\ of\ Topkapi}$$

(a) Performance comparison with *Exact Method -*
`Spark wordcount() + parallel_sort()` for
16GB and 128GB data. Number of threads per
node is 8. Used a cluster of Intel®Westmere processors with each node having 12 cores.

(b) Performance comparison with *Exact Method -*
`Spark wordcount() + parallel_sort()` for
varying data size on 8 nodes. Number of threads
per node is 8. Used a cluster of Intel®Westmere
processors with each node having 12 cores.

Figure 1: Performance results for comparison with `Spark wordcount() + parallel_sort()`

## 5.2 Datasets

We give a thorough performance evaluation on standard large-scale word counting benchmarks
evaluating all possible aspects of the algorithms. We used two sources to compose our data of
different sizes:

**Gutenberg -** This is text data from the Project Gutenberg [1] corpus. The data consists of text from
eBooks in the English language. The data used in our experiments of size up to 16GB are from this
source.

**PUMA Dataset -** We also used "Dataset2" of size 150GB under description "Wikipedia" from
PUMA Datasets [9]. We created data of size 32GB, 64GB, and 128GB from this data set to use in
our experiments.

The task is to identify the top-100 most frequent words in the data, i.e. we use **K=100** for all the
experiments unless otherwise stated explicitly.

## 6 Performance Comparison with Exact Method

Here, we compare the performance of *Topkapi* against "exact methods" which give completely
accurate results at a cost of linear memory space and communication. Representative from this class
of algorithms, we select the popular *Spark wordcount() + parallel_sort()* method.

### 6.0.1 Scalability over Number of Nodes

We present strong scaling (fixed data size) performance results over varying number of nodes for two
different data sizes: a) 16GB (Gutenberg dataset) and b) 128GB (Puma dataset). Figure 1a gives the
overview of speedup variation of *Topkapi* over *Spark wordcount() + parallel_sort()* method for 1
to 16 nodes with each node running 8 threads. As expected, we see significant speedups across the
board. *Topkapi* gives 8x-20x speedup over *Spark wordcount() + parallel_sort()* method for both
the data sizes. In this case, the costly sorting step associated with the exact method incurs a huge
performance penalty.

### 6.0.2 Scalability over Data Size

To see the effects of data size on performance, we fix the number of nodes to 8 and vary the data size
from 16GB to 128GB. The resulting plot with speedup over *Spark wordcount() + parallel_sort()* is
given in Figure 1b which represent 10x-15x speedup over *Spark wordcount() + parallel_sort()*.