[Reviews · NeurIPS 2018]

Reviewer 1



This submission proposes a new technique for Top-K frequent elements (aka "heavy hitters") problem, with a particular focus on practical implementation concerns of approximation accuracy, update speed, and suitability for both single-machine and distributed parallelism. The crux of the idea is to embed the "Frequent" algorithm as a subroutine into the Count-Min-Sketch (CMS) approach in order to avoid the update expense and inaccuracy of the "candidate set" auxiliary heap data structure. The theoretical results show good approximation, and the experimental results show good accuracy and performance in practice. The paper is clearly written, the experimental setup is thoroughly described, and the technical details seem sound (I read the supplement but did not work through the proofs). It seems that this approach could find usefulness in practical applications, but it is not obvious to me that it will have a large research impact. It would strengthen the work to further explore or explain how and why the gains of this approach will influence future research directions. One possible improvement is to compare against baselines other than Frequent (2003) and CMS (2005) - it seems unlikely there have not been other variations or extensions in this problem domain targeting the distributed setting? L210: the text mentions "the risk of slightly more approximation" due to using LHHcounts instead of the standard CMW counter but I didn't see anything in the main submission (or supplement) quantifying this? In general for the setting considered here are we concerned primarily with identifying the correct Top K, or also with the accuracy of their estimated frequencies as well? L305: another naive baseline might be to simply do CMS with heaps of size C*Kfor some constant factor C (eg, C=3). UPDATE: I have read the author response and found it useful. Ideally the final submission could incorporate some of this info, in particular question of whether the problem setting is "Identify Top K elements" vs the actual counts (which Reviewer #2 also found confusing) and the clarification about the 4K heap size for the experiments.

Reviewer 2



This paper presents topkapi, an new algorithm that estimates the most frequent elements in a set in a distributed fashion. This paper presents some positive developments on an important topic for big data and would be a good candidate for publication at this conference. However, this reviewer feels that the paper could be significantly strengthened before publication. - structure: the current paper goes at length to describe a hybrid between CMS and FA, but it casually turns around in section 3.3 to essentially dismiss the sketch part of the algorithm. The reviewer would suggest to focus earlier on presenting the algorithm as FA with sketch-flavored hashing. - proofs: this reviewer is of the opinion that formal guarantees are important for such algorithms and should be made explicit to practitioners. Unfortunately, all the theorems are relegated to the appendix. At least presenting the main results in the paper would reinforce the paper, as it currently is presented more as a discussion without obvious formal backing. - presentation: CMS computes counts of elements, while FA focuses on the identification of the most frequent elements. The paper puts them on equal footing, which led this reviewer to wonder if identifying or computing the frequencies of the top-k elements was the eventual goal of the paper. In particular, presenting CMS as not good for reducibility seems to distract the flow, as pure sketches (without tracking the top elements in a heap) are excellent for distribution and reducibility. Some other comments: - Both 'Frequent' and 'FA' are used, which is a bit confusing. The authors should pick one notation. - The reviewer wonders how the guarantees would be strengthened if, for each of the CMS bins in the sketch, 2 or 3 FA counts would be kept instead of just one. The update would still be trivial with 2 comparisons, but it could lead to more robust estimates, at least in theory. - In practice, it is useful to also have access to the frequencies of the most frequent elements (which is provided by sketches). How would the authors plan to address this point? - Regarding the experiments, the reviewer would be curious to see how pure CMS fares (with a first sketch building pass followed by a heavy hitter hash-based identification pass). This would be more fair and also simpler than a heap augmentation of CMS. Overall, the reviewer would welcome this contribution, provided the points above are addressed. UPDATE: The reviewer thanks the authors for their comments. - regarding the 2-pass CMS + Top-K, this is a 'natural' approach for batch or memory-constrained settings (and of course prohibitive for streams, but the context of the paper is more general than streaming). Given the good performance of topkapi, I would recommend this is included in the final submission, especially as it explicitly now compares against spark, and spark has both built-in CMS and top-k. - regarding bounds on results, the authors rightfully mentioned that CMS and FA provide upper and lower bounds on the exact value. I believe this is worth mentioning as this is important information in practice - the updated version of the paper mentions spark prominently in the abstract as a point of comparison, but the reviewer sees no experiment against it (although the results would be expected to be quite in favor of topkapi). This should be addressed in the final version, either by removing mention of spark or at least mentioning some experiments. These are fairly minor points though, and this reviewer would welcome this contribution provided the points above are addressed in the final revision.

Reviewer 3



This paper targets the problem of identifying top-K frequent items in a dataset, and presents Topkapi, a new algorithm combining the pros of two conventional algorithms (Count-Min Sketch and Frequent) while avoiding the cons carefully. The illustration of the proposed idea is clear and straightforward, and the evaluation shows its advantages over existing algorithms. However, I think the paper needs further improvement: - Some details should be presented in the paper rather than in the supplementary material, e.g., the key properties of Count-Min Sketch and Frequent, the *K* used for evaluation, etc. Also, the abstract mentioned the speedups over Spark but related results are not included in the paper but given in the supplementary material. - Each cluster node has 12 cores but the experiments used only 8 of them. Why not using all of them? Please explain. - The experiments only tested top-100. What would the results be when K varies (i.e. smaller and larger)? As the algorithms (and the way to parallelize computations) look cache-sensitive, the performance of compared implementations could be very different. - What about other variations of Count-Min Sketch, e.g. discarding the heap and re-scan the dataset to identify top frequent items? - Some related work is not discussed, e.g. Efficient Frequent Item Counting in Multi-Core Hardware [KDD'12]. UPDATES: According to the response, it turns out that the proposed and compared algorithms are all cache sensitive (at least in the conducted experiments), so it would be better if we can see the impacts of K in a larger span e.g. 10, 50, 100, 200, 1000, etc. Some profiling results (e.g. the number of instructions, memory accesses, cache misses for each algorithm, etc.) could help.

Reviewer 4



This paper proposes a novel algorithm for find Top-K frequent elements, the algorithm is based on the well-known Count-Min Sketch method and combine the Frequent based counting algorithm for better reducibility and parallelism. The authors also provide good theoretical results on their algorithm. Personally I think this paper is well-written and solving an important problem, however the major reason of me giving a weak rejection is that the experimental results seem to be not finished, the figure 1a - 1e only compare the CMS and FA methods and no Topkapi related results is reported. And to be honest these figures are extremely difficult to see because the font size is too small, and in figure 1f, I just cannot tell which part is the "Merging Thread Local Summaries". UPDATES: The authors did clarify the results reported in the figures, these results are able to demonstrate the parallelism in the Top-K frequent problem is improved with proposed algorithm. Minor Comments: - In line 107, over $D_1 \cup D_2$ should be $S_1 \cup S_2$. - There are no definition for $\epsilon$ and $\delta$, although their meanings can be inferred from the references.